# Successive Paradigm Shifts in the Bacterial Cell Cycle and Related Subjects

**DOI:** 10.3390/life9010027

**Published:** 2019-03-07

**Authors:** Vic Norris

**Affiliations:** Laboratory of Microbiology Signals and Microenvironment, University of Rouen, 76821 Mont Saint Aignan, France; victor.norris@univ-rouen.fr

**Keywords:** philosophy of science, Kuhn, Fleck, DNA replication, cell division, hyperstructure, phenotypic diversity, heterogeneity, metabolon

## Abstract

A paradigm shift in one field can trigger paradigm shifts in other fields. This is illustrated by the paradigm shifts that have occurred in bacterial physiology following the discoveries that bacteria are not unstructured, that the bacterial cell cycle is not controlled by the dynamics of peptidoglycan, and that the growth rates of bacteria in the same steady-state population are not at all the same. These paradigm shifts are having an effect on longstanding hypotheses about the regulation of the bacterial cell cycle, which appear increasingly to be inadequate. I argue that, just as one earthquake can trigger others, an imminent paradigm shift in the regulation of the bacterial cell cycle will have repercussions or “paradigm quakes” on hypotheses about the origins of life and about the regulation of the eukaryotic cell cycle.

Scientific fields sometimes undergo "paradigm shifts" that, according to Kuhn, allow the scientific community to explore new approaches within a fundamentally new view of the world, termed a new paradigm [1]. Paradigm shifts are usually resisted by the community, which sees no need to disrupt the logical and empirical structure of the old paradigm, but which is eventually forced to undergo a shift due to overwhelming inconsistencies between the old paradigm and reality. Progress after a shift is rapid but eventually slows down as the new paradigm is explored. When new contradictions and conflicts with observation arise during that exploration, a new shift is needed. Fleck claimed that, "Once a structurally complete and closed system of opinions consisting of many details and relationships has been formed, it offers enduring resistance to anything that contradicts it." [2]; that is to say, whatever runs counter to the collective way of thinking (i.e., the current paradigm) risks being unseen, ignored, kept secret, or explained away. It might be argued that this makes sense in light of an evolutionary struggle for survival between tribes, where selection probably favored individuals who agreed readily with one another because this allowed their tribe to behave coherently. In the hypotheses of “competitive coherence” [3] and “shared reality” [4], the result is a pressure to share beliefs that is as powerful in science as in religion, politics, and economics. In other words, a paradigm can be adopted for scientific reasons but maintained for non-scientific ones.

For a scientific community, a paradigm shift is about as unwelcome as an earthquake, and just as an earthquake can be preceded by foreshocks and followed by aftershocks, a paradigm shift in one subject can be preceded or followed by shifts in neighboring subjects. An example of a shift is the long period taken to shift from proteins to DNA as the determinants of heredity, a period that lasted from Griffith’s genetic experiments on transformation in 1928, through the Avery, Macleod, and McCarty experiments in 1944 on the chemical nature of the transforming principle [5], to the Hershey and Chase experiments in 1952 [6]. This shift was followed by very rapid progress in many related fields. Another less discussed paradigm shift began only a few decades ago. Until then, it was widely believed that bacteria were simple and lacked the high degree of internal structuring characteristic of eukaryotic cells. It was also believed that the bacterial cell cycle depended on the zonal growth of the peptidoglycan wall to segregate daughter chromosomes and to drive cell division. Despite the reservations of many in the cell cycle field about this paradigm [7], it proved difficult to shift. Now, we know that bacteria are highly structured. Molecules and macromolecules often come together into spatially extended assemblies with specific functions. These assemblies include cytoskeletal filaments resembling actin [8], tubulin [9], and, in some cases, intermediate filaments [10], which all play major roles in the cell cycle events of segregation [11] and division [12]. Such assemblies also include RNases [13], EF-Tu [14,15], CTP synthase [16], microcompartments for sequestering metabolic intermediates [17], “nucleoli” [18,19], chemoreceptor arrays [20], acidocalcisomes [21], clusters of the E1 protein of the phosphotransferase system [22], and associations of genes, nascent RNA, and nascent proteins formed by the coupling of transcription, translation, and either insertion into membrane or into cytoplasmic complexes [23,24] (for additional references see [25]). These widely varying types of assembly, which have been termed hyperstructures, often have different dynamics and may be classed tentatively into non-equilibrium hyperstructures appropriate for growth and equilibrium hyperstructures appropriate for survival [26]. In view of the importance of coherent diversity (see below), it should be noted that having a small number of hyperstructures per cell rather than a host of enzymes permits a greater diversity of coherent phenotypes in the population. 

Paradigm shifts are considered to be brutal and to progress rapidly. However, the shift away from the paradigm of the bacterial cell as an unstructured container has not occurred brusquely, but gradually. There are at least two reasons for this—one is that the complex richness of biology often allows a paradigm to be defended in many ways, for example, by the addition of new regulatory loops (much as the Ptolemaic model of the Universe was saved by the addition of epicycles); the other is that it has taken many years for the many experimentalists involved to accumulate the quantity of results needed. Hence, the shift away from the unstructured cell paradigm has been more of a slippage than a rupture, which may be why it has received little comment.

Following the shift away from the paradigm of unstructured bacteria controlled by the dynamics of the peptidoglycan envelope, another related paradigm has begun to shift. This second paradigm concerns the bacterial cell cycle where two essentially pragmatic assumptions have been made. It has long been assumed that for bacteria (e.g., *Escherichia coli*), the growth rate of the average cell represents the growth rate of all the cells in the population, provided that these cells are growing in steady-state, a condition in which the values of extensive properties, such as the size distribution and the DNA:mass ratio, remain unchanged over time [27,28,29]. It is also assumed that a cell divides to yield daughter cells that are very similar to one another and to their mother at the same age. In other words, growth rates and other cell cycle parameters can be determined by studying aggregates of cells rather than single cells [30]. This “average cell” paradigm is supported by some evidence that each individual bacterium grows at the same exponential rate [27,31], despite some controversy over whether this growth is really exponential [32,33]. This paradigm has permitted major advances in our understanding of the bacterial cell cycle such as its subdivision into B, C, and D periods (for the time between birth and the initiation of DNA replication, the time taken for DNA replication, and the time between the termination of DNA replication and cell division, respectively) [34,35,36]. The paradigm has also permitted the influential hypothesis of an “initiation mass” since, at different growth rates of the population, the ratio of the mass of the average cell to the number of its origins of replication at the time of initiation could be estimated to be constant [37,38]. This finding underpins attractively simple hypotheses about the accumulation of activators (or the dilution of repressors) of initiation, as cell mass increased until the critical initiation mass was reached [37,39]. In *E. coli* and many other bacteria, this initiator of replication is often considered to be the DnaA protein [40,41,42]. 

The above changes in understanding have implications for the behavior of cells that have not yet been widely understood. If a cell is considered as an undifferentiated container of solutes, then the concept of the “average cell” makes sense. Indeed, the average properties of populations are extremely valuable, and from the average properties of molecules to galaxies, using population statistics allows good scientific research to be conducted. The fact that there is variation within a cell population around that average has been acknowledged for decades. However, if the cell is considered as a container of a very small number of large structures, whether they be complex assemblies or microscopically recognizable organelles, then we must question what “average cell” means, and this is particularly important in the context of growth rates. One of the little-appreciated consequences of the paradigm shift from “bacterial cell as droplet” to “bacterial cell as structured entity” is the effect on the paradigm of the cell cycle based on the average cell. The idea that the average cell is sufficient to represent the population is at variance with recent evidence for widespread growth rate diversity, even if it could be contended that these cultures were not in steady-state growth. Video-microscopy has shown that individual cells of *E. coli* can grow in the same conditions with different rates [43] and that such growth rates may even vary by a factor of over four [44,45,46]. A very different high-precision technique, which depends on determining the buoyant densities of individual *E. coli*, has also shown that their growth rates can vary widely [47]. By combining labeling with stable isotopes and secondary ion mass spectrometry, we have shown that the growth rates of individual *E. coli* vary four-fold in liquid media, even in samples of a couple of hundred cells [48]. The point is that there is a big difference between believing that a distribution of the values of a global parameter, such as growth rate in a population of non-differentiating bacteria, is unavoidable but relatively unimportant, and believing that this distribution is regulated and fundamental. It is in this changing belief that the paradigm shift is occurring.

The paradigm of the “initiation mass” has also taken a few knocks. Simple models based on the dilution of a repressor or the accumulation of an activator, with respect to an origin of replication, fail to explain how cells not only maintain minichromosomes (small plasmids that replicate using the chromosomal origin of replication) [49], but also replicate scores of them in synchrony with the chromosome [50]. Moreover, it has been claimed that the initiation mass varies at different growth rates, and that there is no direct connection between mass accumulation and the molecule(s) determining initiation of replication [51,52,53] (but see also [54]). Most serious for simple models, the presence of an additional copy of *oriC* in the *E. coli* chromosome, which results in two origins functioning at the same time, has little effect on the cell cycle [55,56]. Other criticisms have also been leveled at the accumulation of the DnaA protein as the key process in initiation [57] and at the “initiator mass” paradigm itself [58]. If initiation is the first event in the cell cycle, the final event is division. Here there is uncertainty with the positioning of the divisome, which is no longer believed to rely entirely on mechanisms based on Min, SlmA, and the putative Ter macrodomain-FtsZ-ring link [59].

The fact that bacteria sometimes have different phenotypes (and, in particular, have different growth rates) in the same media in the same conditions means that there is a need to move away from the old cell cycle paradigms of “average cell” and “initiation mass”, and towards a new paradigm. One way to find this paradigm is to go back to basics and ask the question, "What is the problem?". A fundamental problem for biological systems at all levels is how to adapt to changing environments. An evident solution for cells (and arguably other systems) lies in phenotypic diversity. A subset of this problem is how to reconcile the incompatible strategies needed for growth and survival. In an uncertain environment, cells may sometimes need to grow rapidly to outpace competitors or to grow very slowly (or not at all) to avoid danger (e.g., from nutrient depletion and antibiotics). Here, the solution again lies in a diversity of phenotypes, with a cell being imagined as orbiting in a vast “phenotype space” around the two attractors of growth and survival. The question then is how to generate phenotypes that are (1) internally coherent so that an individual bacterium has functions that are only compatible with one another and (2) externally coherent so that the distribution of phenotypes within a population is the best compromise for growth and survival, which has been termed “life on the scales of equilibria” [58]. This latter requirement can be satisfied if there is a coherence in the successive generations of bacteria (i.e., in the coherence of trajectories in phenotype space). It has been argued that this question of how to generate coherence in phenotypes should lead to a candidate for a new paradigm for the cell cycle [58].

This new paradigm could be based on the hypothesis that the cell cycle itself is responsible for generating coherent phenotypic diversity. In support of this diversity hypothesis, the segregation of intracellular material in *E. coli* is asymmetric [44,60,61], which, followed by cell division, could generate a coherent metabolic diversity if different sets of hyperstructures were to accompany each of the parental DNA strands [62,63]. Such metabolic diversity could also result from variations in the speed of replication in different regions of the chromosome, again followed by cell division [64]. Consistent with all this is the importance of cell division in determining diversity and growth rates in bacteria [65,66,67]. In the diversity hypothesis, the timing of initiation of replication depends on intracellular structures and, in particular, the state of hyperstructures, which determines where a bacterium is in phenotype space and where its daughters will be. The idea is that bacteria initiate DNA replication in response to the values of two parameters: (1) the quantity of survival-related, equilibrium hyperstructures that have accumulated (which might be considered a variant of the “adder” model [46,68,69,70,71]) and (2) the intensity with which their growth-related, non-equilibrium hyperstructures are functioning [58,63,72]. The diversity hypothesis helps explain not only the relationship between intracellular structure, phenotypic diversity, and the cell cycle but also why growth rate diversity is sometimes not found [31]. In the latter experiments, the mass doubling times of the bacteria were less than the time needed to replicate the chromosome, resulting in the cells always containing more than one copy of the chromosome (sometimes many copies) [73]; this could result in an averaging within the cell―a convergence on the mean―of what would otherwise be different growth rates if the chromosomes were in separate cells such as during slow growth.

If the diversity hypothesis were to become the new cell cycle paradigm for bacteria, it might have an impact on other research fields. For example, it would strengthen the hypothesis of the prebiotic ecology in which life originated in an abiotic flux in which a huge variety of molecules were created (or arrived from outside the system) and were then destroyed [74]. In this flux, those molecules that interacted with one another due to their complementary structures or “molecular complementarity” were preserved from destruction and therefore accumulated in the form of a rich, diverse population of composomes that displayed “compositional inheritance” insofar as the compositions of the daughters of a composome resembled the composition of the parent composome [75,76]. Selection in the prebiotic ecology is at the level of a diverse population as it is in the diversity hypothesis. The fusion and fission of composomes [77] has its modern equivalent in the dynamics of hyperstructures, again underpinned by molecular complementarity. The fission that generated coherent diversity in the population of composomes—for example, by separating autocatalytic networks that might interfere with one another [78]—has its equivalent in the division that can separate equilibrium and non-equilibrium hyperstructures so as to generate coherent phenotypic diversity in populations of modern bacteria [48].

The echoes of the prebiotic ecology in the modern world can also be detected in the structural integrity of cells. The combination of the elastic properties of DNA and the coupled transcription–translation–insertion (transertion) of peptides into membranes may have contributed to the structural integrity of protocells [79], just as transertion is believed to contribute to the integrity of modern cells [80]. It is proposed that the association between extended structures and catalysis that occurred at an early stage in the prebiotic ecology [81] has an echo in the *enzoskeleton* of modern bacteria [16,82]. Another aspect to structural integrity is the problem of maintaining it when a system grows. In the case of cells or protocells, this problem occurs because the average connectivity between cellular constituents per unit mass decreases as a system grows; in other words, growth puts identity at risk [63]. One proposed solution to restoring connectivity in growing cells is to initiate DNA replication and then go through the cell cycle. The timing of initiation of replication in bacteria in our intensity-sensing model depends on the increasing activity of one of the most important hyperstructures, the ribosomal hyperstructure or bacterial “nucleolus” [58]. Correspondingly, on the basis of sequence similarities between rRNA and the mRNA encoding ribosomal proteins, ribosomes have been accorded the primary role in the transition from the abiotic world to the first cells [83].

In the diversity hypothesis, the existence of many hyperstructures depends on the molecular complementarity between what we term simple, universal molecules and ions (SUMIs). SUMIs include polyphosphate, poly-(R)-3-hydroxybutyrate, polyamines, and small inorganic ions such as phosphate [84]. We propose such interactions create coherent behavior in modern cells just as they created it in the prebiotic ecology. In modern cells, for example, at one level the SUMIs interact with one another to make voltage-gated calcium channels in the case of polyphosphate binding to poly-(R)-3-hydroxybutyrate [85], and at a higher level they interact with one another to make hyperstructures such as acidocalcisomes [86]. The SUMIs and the hyperstructures they make are part of the production in the prebiotic ecology of interconnected metabolic pathways that arose simultaneously with phosphate, which is of central importance in the origins of life [87,88], and can catalyze the simultaneous production through a common chemistry of the precursors of ribonucleotides, amino acids, and lipids [89].

There is no consensus as to why eukaryotic organelles have their own DNA. If the diversity hypothesis was to become the new cell cycle paradigm, it might strengthen the proposal that mitochondria possess DNA in order to have hyperstructures that structure their membranes and cytoplasm and allow their cell cycles to generate phenotypically different mitochondria [90]. There is an increasing body of evidence for hyperstructures in mitochondria, including those for the oxidative phosphorylation system [91], for nucleoid structure [92], for RNA processing, and for ribosome assembly [93,94,95]. The evidence that mitochondria are indeed phenotypically different from one another and that these differences are intimately related to the replication of their DNA [96,97] has also increased.

In this context, given the similarities and evolutionary relationships of prokaryotic and eukaryotic cells, the aftershocks of a paradigm shift for the bacterial cell cycle might help shift yet another paradigm, namely, the cyclin- and checkpoint-based paradigm for eukaryotes, which is not without its fracture lines [98,99,100]. One example of this is the restriction point, a unique point in the cell cycle where cells come to rest without entering S-phase when their growth is arrested by starvation or inhibition [101]. However, there is a question regarding this interpretation given the results from bacteria grown in different media where limitation caused cells to begin or not begin DNA replication at different times [100]. Another question regards the significance given to results that depend on “synchronizing” a population by treatments that block entry to S-phase. It has been pointed out that such treatments produce a population of cells that differ greatly since a cell that has just started to replicate its DNA will go on to divide, while a cell that was about to replicate its DNA will not go on to divide. Hence, the former cell could be half the size of the latter, a variability that opens the door to misinterpretations if this population is considered to be synchronized [98]. A third question regards the cyclin-CDK paradigm given that a cyclin-CDK chimera can substitute for the four cyclin-CDK complexes acting during the mitotic cell cycle and the six cyclin-CDK complexes acting during the meiotic cell cycle [99]. This question has led to criticisms and to a revised model [102]. One might ask why such a system should have evolved at all, why it should be based on phosphorylation when there are other―perhaps more fundamental―post-translational modifications available [103], why it fails to shed a light on other problems in biology, and whether it might just be a molecular veneer overlying a deeper level of control. It is worth recalling here that the paradigm that bacteria had no protein kinases phosphorylating on serine, threonine, and tyrosine residues has shifted [104,105,106,107].

A significant brake on scientific progress stems from the failure to recognize and to counteract the non-scientific reasons that make a paradigm hard to shift. The consequences of maintaining a paradigm for non-scientific reasons are multiple. A recent analysis has shown that most research concentrates on a minority of genes and proteins due to factors that are unrelated to their functional importance [108]. These factors include the perceived risk to careers incurred by studying genes that few others have studied (i.e., incurred by being an outsider who may be seen as not subscribing to the paradigm). Another comprehensive analysis of citations has shown that, as the annual number of publications increases, scientific fields become ossified, which leads the authors to ask “Could we be missing fertile new paradigms because we are locked into over-worked areas of study?” [109]—in other words, locked into a paradigm. Part of the solution is to resist the tendency of a popular hypothesis to become a paradigm because of the danger that this represents. Our preference for black or white (true or false) classifications facilitates this tendency. A preference for a chromatic scale would be better, with the position on the scale reflecting the credibility of the hypothesis at a particular time. Furthermore, it should be accepted that this position would vary over the scientific community and over time. The generation of phenotypic diversity is of evolutionary advantage to many species of bacteria. We might learn from them and consider each hypothesis as an individual bacterium in a population in which they grow, differentiate, sporulate, mutate, conjugate, and compete. An example of a hypothesis “sporulating” is that of the helical clock, which offers an explanation for how cell cycle events are directed and executed [110,111]. Although now largely ignored, a form of this hypothesis resurfaced (alias ‘germinated’?) with the investigation of the spiral patterns of the actin-like proteins in bacteria [112].

In summary, the hypothesis presented above is that a paradigm shift in one subject can lead to paradigm shifts in related subjects. The evidence for this is that (1) there has been a paradigm shift away from the view of bacteria as unstructured; (2) there is, in the case of non-differentiating bacteria, a paradigm shift underway from the growth rate of the average bacterium to the growth rates of individual bacteria; and (3) there are the precursors characteristic of a paradigm shift in our understanding of the bacterial cell cycle. The hypothesis makes the testable predictions that a prebiotic ecology of composomes will become the new origins of life paradigm and that there will be a shift away from the current paradigm for the eukaryotic cell cycle.

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
