# Peer review of "Successive Paradigm Shifts in the Bacterial Cell Cycle and Related Subjects"

_life, 2019, doi:10.3390/life9010027_

Round 1
Reviewer 1 Report
This paper proposes that a paradigm shift in one field can lead to paradigm shifts in related fields. This is a very important idea, from a theoretical systemic view point It also examplify this idea by using the example of the paradigm shift in the field of the intracellular structure of bacteria to illustrate how this is affecting the paradigm of the bacterial cell cycle based on the average cell (in which phenotypic diversity is effectively absent) and the related paradigm of initiation mass based on the accumulation of the DnaA protein. The author then proposes a new paradigm for the cell cycle based on what he claims is its fundamental role in generating phenotypic diversity. It is argued that if this diversity hypothesis were to become the new paradigm, it would also affect the status of a related idea in the field of the origins of life and might even affect the current cyclin paradigm of the eukaryotic cell cycle.
This paper is something of a mixed bag of established facts and questionable speculations. The introduction to the work of Ludwik Fleck and the concept of Shared Reality, which many readers may not know, is interesting. It is true that the general view of bacteria as unstructured sacks of freely diffusible molecules and macromolecules has long been abandoned (though few authors have adopted the term hyperstructures). There is nothing new in the idea that division can generate bacteria with different phenotypes – bacterial differentiation is evident to all microbiologists; that said, the subjects are often compartmentalised as may be the case for the non-differentiating E. coli.
The idea that steady state growth is unattainable is unconvincing: a bacterium with 2000 ribosomes and a bacterium with 2001 ribosomes have effectively the same phenotype; steady state growth remains an extremely useful concept. The speculations about life on the scales, intensity and quantity sensing, and division as a way to restore connectivity will be unfamiliar to many readers (even if they have been published). Even though they are intriguing, they may not be found plausible. The introduction of these speculations does not therefore help make a compelling case for the idea that I think very important, that a shift in one paradigm might lead to a shift in another.
It is not clear that the molecular complementarity between polyphosphate and PHB means that these two molecules actually bind to one another to create ion channels. Nor is it clear that there are fault lines in the field of the eukaryotic cell cycle.
Author Response
REVIEWER 1
“The idea that steady state growth is unattainable is unconvincing: a bacterium with 2000 ribosomes and a bacterium with 2001 ribosomes have effectively the same phenotype; steady state growth remains an extremely useful concept.” I have now deleted the entire section (see above).
“The speculations about life on the scales, intensity and quantity sensing, and division as a way to restore connectivity will be unfamiliar to many readers (even if they have been published). Even though they are intriguing, they may not be found plausible.” Even if I fail to convince readers of the plausibility of these speculations they may be helpful as illustrations of the type of paradigm that the field may soon adopt.
“It is not clear that the molecular complementarity between polyphosphate and PHB means that these two molecules actually bind to one another to create ion channels.” They do indeed bind one another in an in vitro system to create ion channels. We have therefore made this explicit “to make, for example, voltage-gated calcium channels in the case of polyphosphate binding to poly-(R)-3-hydroxybutyrate”
“Nor is it clear that there are fault lines in the field of the eukaryotic cell cycle.” We have expanded this paragraph to make the argument clearer: “In this context, given the similarities and evolutionary relationships of prokaryotic and eukaryotic cells, the aftershocks of a paradigm shift for the bacterial cell cycle might help shift yet another paradigm, namely, the cyclin- and checkpoint-based paradigm for eukaryotes, which is not without its fracture lines [98-100]. One example of this is the restriction point, a unique point in the cell cycle where cells come to rest without entering S-phase when their growth is arrested by starvation or inhibition [101]. A question mark, however, lies over this interpretation given the results from bacteria grown in different media where limitation caused cells to begin or not begin DNA replication at different times [100]. Another question mark lies over the significance given to results that depend on ‘synchronising’ a population by treatments that block entry to S-phase; it has been pointed out that such treatments produce a population of cells that differ greatly since a cell that has just started to replicate its DNA will go on to divide whilst a cell that was about to replicate its DNA will not go on to divide; hence, the former cell could be half the size of the latter, a variability that opens the door to misinterpretations if this population is considered to be synchronised [98]. A third question mark lies over the cyclin-CDK paradigm given that a cyclin-CDK chimera can substitute for the four cyclin-CDK complexes acting during the mitotic cell cycle and the six cyclin-CDK complexes acting during the meiotic cell cycle [99]; this has led to criticisms and to a revised model [102]. One might ask why such a system should have evolved at all, why it should be based on phosphorylation when there are other – perhaps more fundamental – post-translational modifications available [103], why it fails to shed a light on other problems in biology, and whether it might just be a molecular veneer overlying a deeper level of control. It is worth recalling here that the paradigm that bacteria had no protein kinases phosphorylating on serine, threonine and tyrosine residues has shifted [104-107].”

Reviewer 2 Report
See attached file

Author Response
Please see the attachment for responses to reviewer 2
